# HIDISC: A Hyperbolic Framework for Domain Generalization with Generalized Category Discovery

**Vaibhav Rathore**[1]    **Divyam Gupta**[1]    **Biplab Banerjee**[1]
Indian Institute of Technology Bombay[1]
{vaibhav.rathor.in, divs25.iitb, getbiplab}@gmail.com

## Abstract

Generalized Category Discovery (GCD) aims to classify test-time samples into either seen categories—available during training—or novel ones, without relying on label supervision. Most existing GCD methods assume simultaneous access to labeled and unlabeled data during training and arising from the same domain, limiting applicability in open-world scenarios involving distribution shifts. Domain Generalization with GCD (DG-GCD) lifts this constraint by requiring models to generalize to unseen domains containing novel categories, without accessing target-domain data during training. The only prior DG-GCD method, DG$^2$CD-Net [1], relies on episodic training with multiple synthetic domains and task vector aggregation, incurring high computational cost and error accumulation. We propose HIDISC, a hyperbolic representation learning framework that achieves domain and category-level generalization without episodic simulation. To expose the model to minimal but diverse domain variations, we augment the source domain using GPT-guided diffusion, avoiding overfitting while maintaining efficiency. To structure the representation space, we introduce *Tangent CutMix*, a curvature-aware interpolation that synthesizes pseudo-novel samples in tangent space, preserving manifold consistency. A unified loss—combining penalized Busemann alignment, hybrid hyperbolic contrastive regularization, and adaptive outlier repulsion—facilitates compact, semantically structured embeddings. A learnable curvature parameter further adapts the geometry to dataset complexity. HIDISC achieves state-of-the-art results on PACS [2], Office-Home [3], and DomainNet [4], consistently outperforming the existing Euclidean and hyperbolic (DG)-GCD baselines.[1]

## 1 Introduction

Deep neural networks have achieved impressive success in visual recognition [5, 6], yet typically assume a shared domain and label space between training and test data. This assumption breaks down in real-world applications such as autonomous driving [7] and medical diagnostics [8], where both *domain shift* and *label shift* frequently co-occur. While semi/self-supervised learning [9, 10] reduces labeling demands, it still operates under closed-world constraints. Domain Adaptation (DA) and Domain Generalization (DG) [11, 12] address distribution shift but assume a fixed set of categories. Open-set DG [13, 14] allows test-time novelty but collapses all unknowns into a single rejection class, erasing semantic granularity.

Generalized Category Discovery (GCD) [15–17] seeks to identify both known and novel classes from unlabeled test data but requires joint access to labeled and unlabeled samples from the same domain. Cross-Domain GCD (CD-GCD) [18, 19] introduces domain shift but still assumes concurrent access to source–target domains during training. In contrast, **Domain Generalization with GCD (DG-**

---

[1]Code Link : https://vaibhavrathore1999.github.io/HiDISC/

**GCD)** [1] represents a more realistic setting: *the model is trained solely on labeled source data and must generalize to an unseen target domain containing both seen and novel categories.*

Addressing DG-GCD requires (i) learning domain-invariant features and (ii) discovering novel semantic structures without supervision. The only existing solution tailored for DG-GCD, DG²CD-Net [1] approaches this via episodic training with synthetic domains and task aggregation, but suffers from high computational cost and cumulative approximation errors that limit generalization.

From a different perspective, existing GCD and DG-GCD methods typically rely on Euclidean or hyperspherical geometry [1, 18], which struggle to capture semantic hierarchies. Hyperbolic geometry [20, 21], with its negative curvature and exponential volume growth, offers a natural alternative for modeling inter-class structure (Fig. 1). While hyperbolic embeddings have shown benefits in GCD (HypCD) [22] and DG [23] recently, their use in DG-GCD, where both domain and label shifts co-occur, remains unexplored. This raises our central question:

> *Can hyperbolic geometry provide a unified foundation for solving DG-GCD, addressing both distribution shift and novel-class discovery?*

**Our approach.** We introduce HIDISC, the first hyperbolic geometry-aware framework for DG-GCD that learns semantically structured and domain-invariant embeddings in the Poincaré ball [20] without requiring target supervision. Unlike HypCD [22], which addresses standard GCD in single-domain settings, and DG²CD-Net [1], which operates in Euclidean space and relies on episodic simulation, HIDISC provides a unified, non-episodic solution using the representational advantages of hyperbolic space.

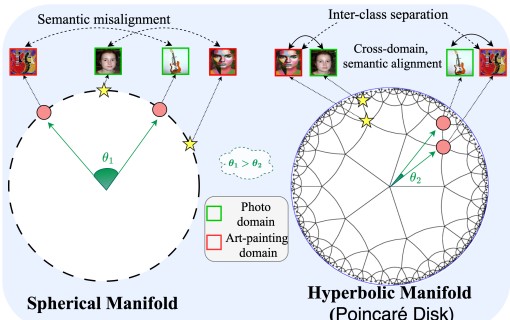

Figure 1: **Spherical** vs. **hyperbolic** (Poincaré) embeddings on PACS. Same-class samples from different domains (green/red) cluster more tightly in hyperbolic space, demonstrating improved class separation. Refer to **Sup. Mat.** for quantitative analysis.

Since the hyperbolic space offers substantial domain invariance and focuses on shared semantics [23] (Fig. 1, **Sup. Mat.**), we still chose to augment the training source domain with controlled stylistic variations and generate only 1–2 synthetic domains per image via a GPT-4o [24]-guided diffusion model, avoiding the computational overhead of task-based simulation in DG²CD-Net. A novel *domain-diversity score* ranks these augmentations by measuring source divergence and intra-pair variability, enabling a principled and scalable domain diversification strategy.

To ensure the model does not overfit the known classes and encourage semantic diversity, we propose *Tangent CutMix*, a curvature-aware interpolation method that mixes labeled features in the tangent space of the Poincaré ball to create pseudo-open samples. Unlike Euclidean mixing used in SimGCD [25] or CMS [26], our method preserves hyperbolic consistency and generates geometrically valid pseudo-novel embeddings that support open-set regularization.

To structure the latent space, we introduce a unified loss that combines three novel components not jointly explored in prior (DG)-GCD literature: (i) a penalized Busemann loss that aligns seen-class features to fixed prototypes at the hyperbolic boundary while reserving interior space for unknowns; (ii) a hybrid hyperbolic contrastive loss balancing angular and geodesic similarities to enable fine-grained clustering across known and novel categories; and (iii) an adaptive outlier rejection loss that pushes synthetic cut-mix samples away from known-class regions, encouraging open-space generalization without relying on adversarial or domain-specific objectives. A learnable curvature parameter further adapts the geometry to dataset-specific complexity. Major contributions include:

– HIDISC, the first hyperbolic DG-GCD framework, jointly handles domain and category shift without target supervision or episodic simulation.

– A unified loss formulation integrating Busemann alignment, hybrid contrastive regularization, and an adaptive outlier repulsion.

– Tangent CutMix, the first open-set augmentation designed specifically for hyperbolic geometry.

– State-of-the-art results on PACS [2], Office-Home [3], and DomainNet [4], outperforming all the baselines consistently and reducing training FLOPs by over $96\times$ vs. DG$^2$CD-Net.

## 2 Related Works

**Domain Generalization.** DG aims to train models on labeled data from one or more source domains to generalize effectively to previously unseen target domains [12, 27]. DG variants include closed-set, open-set, single-source, and multi-source settings [28]. Methods like MixStyle [29] and StyleHallucination [30] enhance robustness via feature-level style perturbations, while meta-learning techniques [31, 32] simulate domain shifts episodically to improve adaptability. *Open-set DG methods address novel test-time classes [33, 34, 14, 35], but typically collapse all unknowns into a single "outlier" class, hindering fine-grained discovery needed in DG-GCD.*

**Category Discovery.** Category Discovery seeks to partition unlabeled data into known and novel categories. While Novel Category Discovery (NCD) [36] assumes complete disjointness between training and test classes, GCD allows overlap and requires identifying both seen and unseen categories during inference [15–17, 25, 37, 38, 26]. Most GCD approaches assume joint access to source and target domains during training, limiting real-world applicability. CD-GCD methods like CDAD-Net [18] and HiLo [28] reduce domain gaps via adversarial alignment or style normalization but still depend on concurrent domain access. To remove this constraint, DG-GCD [1] simulates domain shifts through text-driven image manipulation (e.g., InstructPix2Pix [39]) and aggregates task-specific knowledge via task vectors [40]. *However, these methods operate in Euclidean spaces, which struggle to encode the hierarchical and shared semantic structures crucial for robust domain and category generalization.*

**Hyperbolic Embedding Spaces.** Hyperbolic geometry, defined by negative curvature and exponential volume growth, is well-suited for modeling hierarchical and part-whole semantic structures [20, 21]. Hyperbolic embeddings have improved performance in classification [41–43], few-shot learning [44], segmentation [45], and action recognition [46], supported by hyperbolic variants of standard network components [47, 48, 21, 49]. Recent Busemann-based techniques [23, 50] anchor ideal prototypes on the Poincaré boundary for directional alignment. HypCD [22] successfully applies this to GCD, but assumes joint source–target access. Beyond these, hyperbolic methods have been applied across diverse tasks: Ge et al. [51] explore contrastive learning for hierarchical scene–object representation, Yue et al. [52] study metric learning with hard negatives, Liu et al. [53] extend contrastive learning to EEG, Sun & Ma [54] investigate recommendation, while others address hashing [55] and face anti-spoofing with hierarchical prototypes [56]. *To date, no work has explored hyperbolic representations for DG-GCD, which combines open-set discovery and domain shift without target supervision.*

## 3 Methodology

### 3.1 The DG-GCD Problem Definition

In DG-GCD, we are given a labeled source-domain dataset:

$$D_S = \{(x_i^s, y_i^s)\}_{i=1}^{n_s}, \quad x_i^s \in \mathcal{X}_s, \ y_i^s \in \mathcal{Y}_s,$$

where $x_i^s$ represents source-domain inputs, and $y_i^s$ denotes labels drawn from a set of *known* categories $\mathcal{Y}_s$. At test time, we encounter an unlabeled target-domain dataset:

$$D_T = \{x_j^t\}_{j=1}^{n_t}, \quad x_j^t \in \mathcal{X}_t,$$

where samples belong either to known categories ($\mathcal{Y}_t^{\text{old}} = \mathcal{Y}_s$) or previously unseen, *novel* categories ($\mathcal{Y}_t^{\text{new}}$), such that $\mathcal{Y}_t^{\text{new}} \cap \mathcal{Y}_s = \emptyset$. Crucially, the data distributions across domains differ significantly, i.e., $P(\mathcal{X}_s) \neq P(\mathcal{X}_t)$, and the target dataset $D_T$ is inaccessible during training.

Our objective is to construct an embedding space using only $D_S$ that generalizes across domains and categories, effectively clustering both known and novel-class samples from the unseen dataset $D_T$.

### 3.2 Rationale Behind Using Hyperbolic Space for DG-GCD

Semantic structures in visual data—such as hierarchies, taxonomies, and part–whole relationships—are inherently suited to spaces with exponential capacity. Hyperbolic space, characterized

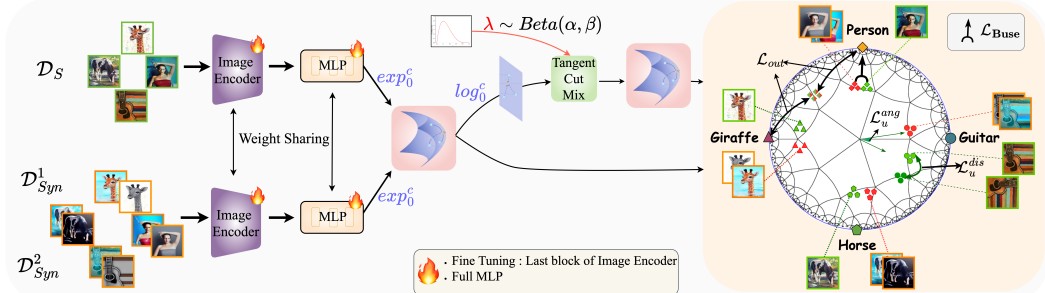

Figure 2: **Illustration of the HıDISC pipeline for DG-GCD in hyperbolic space**. The model is trained using labeled source data $\mathcal{D}_S$ (green borders) and 1–2 GPT-guided synthetic domains $\mathcal{D}_{Syn}^1$, $\mathcal{D}_{Syn}^2$ (orange borders) to simulate domain shift. Features from the shared encoder are projected to the Poincaré ball via $\exp_0^c$. To mimic novel categories, Tangent CutMix performs interpolation in the tangent space and maps the result $z_{mix}$ back to hyperbolic space. The embedding space is structured via: (i) penalized Busemann loss $\mathcal{L}_{Buse}$ for aligning seen classes to boundary-fixed prototypes; (ii) hybrid contrastive loss $\mathcal{L}_u$ for clustering and separability; and (iii) adaptive outlier loss $\mathcal{L}_{out}$ to repel pseudo-novel points. Together, these shape a curvature-aware space for generalization and discovery.

by negative curvature and exponential volume growth, naturally encodes such structures, making it particularly beneficial for DG-GCD, where labeled classes typically reflect coarse semantic strata, while novel categories often reside in finer or more abstract regions.

In contrast to Euclidean or spherical embeddings [16, 57], which are constrained by polynomial growth, hyperbolic embeddings support both local compactness and global semantic separation. Moreover, hyperbolic geometry improves domain generalization by amplifying higher-level semantic distances and attenuating domain-specific low-level variations, thereby fostering robust, domain-invariant representations under substantial distributional shifts (more details provided in **Sup. Mat**).

**Poincaré Ball Geometry.** We adopt the Poincaré ball [20] as our hyperbolic model. For curvature $-c^2$, the $n$-dimensional ball is defined as:

$$\mathbb{D}_c^n = \left\{ \mathbf{a} \in \mathbb{R}^n \mid c\|\mathbf{a}\|^2 < 1 \right\},$$

where $\|\cdot\|$ denotes the Euclidean norm. Additional geometric details are provided in the **Sup. Mat.**

### 3.3 Navigating through HıDISC for DG-GCD

We propose HıDISC (Fig. 2), a hyperbolic framework that jointly addresses the dual challenges of DG-GCD: domain-invariant representation learning and unsupervised semantic disentanglement. The synthesis-driven components of our model include: (i) *Synthetic Domain Augmentation*, which introduces a compact set of diverse, diffusion-generated domains to simulate realistic distribution shifts without relying on target access; and (ii) *Tangent CutMix*, a curvature-aware interpolation mechanism operating in the tangent space of the Poincaré ball, generating pseudo-novel samples while preserving manifold fidelity. Complementing these are three loss-driven modules: (iii) *Prototype Anchoring*, which aligns seen-class embeddings to fixed ideal prototypes on the Poincaré boundary, reserving central space for novel classes; (iv) *Adaptive Outlier Loss*, which ensures synthetic samples are repelled from known-class clusters, promoting open-space regularization; and (v) *Hybrid Hyperbolic Contrastive Loss*, which combines geodesic and angular similarity to improve local cohesion and global separability. Each component is described in detail in the subsequent sections.

#### 3.3.1 Lightweight Synthetic Domain Augmentation

To simulate domain variability without relying on expensive episodic training (as in DG$^2$CD-Net [1]), we generate only one or two synthetic domains per experiment using a diffusion model guided by GPT-4o [24]-curated prompts (e.g., `underwater`, `night-time` variants of *class* instances) (see **Sup. Mat.** for qualitative visualizations). These synthetic domains serve as proxy distributions that expose the model to varied visual shifts and support generalization to unseen domains.

Unlike DG$^2$CD-Net, which depends on numerous episodic tasks and synthetic domain permutations, our strategy is lightweight and avoids both computational overhead and error propagation across

episodes. Crucially, in hyperbolic space, even a small number of diverse augmentations can induce expansive representational changes due to the geometry's exponential capacity—effectively stretching the semantic space and encouraging separation between seen and unseen regions.

First, we introduce a *domain-diversity score* that ranks a given synthetic domain $\mathcal{D}_{\text{syn}}^{(s)}$ with respect to other synthetic domains $\{\mathcal{D}_{\text{syn}}^{(l)}\}_{l=1}^{\mathcal{M}}$ and the source domain $\mathcal{D}_S$ based on the notion of mutual diversity calculated using the Fréchet Inception Distance (FID) [58] for $\mathcal{M}$ synthesized domains:

$$\text{Score}(\mathcal{D}_{syn}^{(s)}) = \frac{1}{\mathcal{M}-1} \sum_{\substack{l=1,\\l\neq s}}^{\mathcal{M}} \left[ \text{FID}(\mathcal{D}_S, \mathcal{D}_{\text{syn}}^{(s)}) + \text{FID}(\mathcal{D}_{\text{syn}}^{(s)}, \mathcal{D}_{\text{syn}}^{(l)}) \right]. \tag{1}$$

This scoring promotes both source-domain divergence and intra-pair complementarity. We select the top-scoring $1-2$ domains to augment $\mathcal{D}_S$ to obtain $\mathcal{D}_{\text{train}}$.

As shown in Fig. 3, excessive augmentation leads to overfitting on seen classes and degrades novel class discovery. *Notably, training solely on $\mathcal{D}_S$ yields competitive results, highlighting the inherent domain robustness of hyperbolic space, and using these augmentation provides marginal boosts.* Effects of redundant augmentations are mentioned in **Sup. Mat.**

### 3.3.2 Mapping Visual Features into Curvature-Aware Hyperbolic Geometry

With the augmented training set, we learn representations using a frozen DINO [59]-pretrained ViT [60] followed by a 3-layer MLP. The resulting Euclidean feature $\mathbf{z}^{\mathbb{E}} \in \mathbb{R}^d$ is projected into the Poincaré ball $\mathbb{D}_c^d$ via the exponential map:

$$\mathbf{z}_i^s = \exp_0^c(\mathbf{z}^{\mathbb{E}}) = \tanh(\sqrt{c}\|\mathbf{z}^{\mathbb{E}}\|) \cdot \frac{\mathbf{z}^{\mathbb{E}}}{\sqrt{c}\|\mathbf{z}^{\mathbb{E}}\|}, \tag{2}$$

where $c$ is a learnable curvature parameter in our case to approximate the data complexity more effectively. Let $z_i := \mathbf{z}_i^s$ denote the hyperbolic feature. This projection facilitates the encoding of hierarchical semantics and ensures geometric consistency in the downstream tasks.

### 3.3.3 Tangent CutMix and Adaptive Outlier Loss

To hallucinate novel-category samples and regularize the open space in hyperbolic geometry, we introduce **Tangent CutMix** [61]—a curvature-aware variant of CutMix tailored for the Poincaré ball. Traditional CutMix interpolates feature representations in Euclidean space to synthesize outliers, which can violate the geometric constraints of hyperbolic space. In contrast, Tangent CutMix performs mixing in the tangent space at the origin, ensuring consistency with the underlying manifold structure.

Given two embeddings $z_i, z_j \in \mathbb{D}_c^d$ with different class labels, we:

(1) **Project to tangent space:** $v_i = \log_0^c(z_i),\ v_j = \log_0^c(z_j)$

(2) **Linear mix:** Compute $v_{\text{mix}}^{i,j} = \lambda v_i + (1-\lambda)v_j$, where $\lambda \sim \text{Beta}(1,1) = \text{Uniform}(0,1)$

(3) **Map back:** $z_{\text{mix}}^{i,j} = \exp_0^c(v_{\text{mix}})$

The resulting embedding $z_{\text{mix}}$ represents a curvature-preserving interpolation of features with incompatible semantics, mimicking out-of-distribution behavior while remaining valid in the hyperbolic space. Furthermore, to prevent these synthetic features from collapsing into known class regions, we apply an adaptive outlier loss:

$$\mathcal{L}_{\text{out}} = \mathbb{E}_{(x,y)\sim\mathcal{P}(\mathcal{D}_{\text{train}})} \sum_{i,j,y_i\neq y_j} \max(0, \gamma - \min_{k\in\mathcal{Y}_s} \mathbb{D}_{\mathbb{H}}(z_{\text{mix}}^{i,j}, \mathbf{p}_k)), \tag{3}$$

where $\mathbb{D}_{\mathbb{H}}$ is the hyperbolic distance, and $\gamma$ is a quantile-based adaptive margin over the distances from all class prototypes in $\{\mathbf{p}_k\}_{k=1}^{|\mathcal{Y}_s|}$. This encourages pseudo-novel embeddings to remain outside the regions occupied by seen classes, effectively reserving space for novel category discovery. For further analysis regarding adaptive margin and the generated CutMix samples, see **Sup. Mat.**

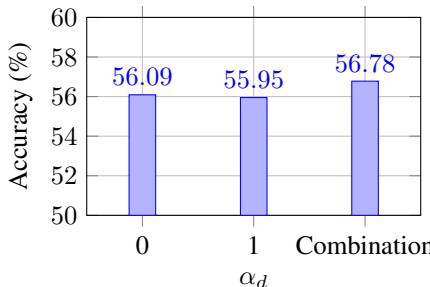 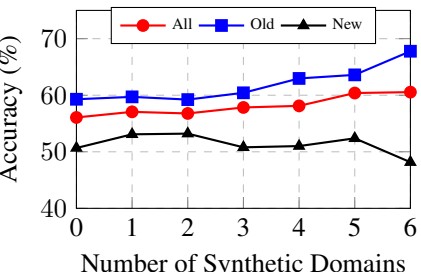

Figure 3: **(Left)** Effect of $\alpha_d$ in hybrid contrastive loss. A balanced combination of angular and geodesic components achieves the highest accuracy. **(Right)** Impact of synthetic domains on old and new category performance. While old-class accuracy increases due to augmented seen data, new-class performance slowly degrades with more synthetic domains, as they cause seen-class bias.

### 3.3.4 Prototype Anchoring with Penalized Busemann Loss

To enforce semantic structure among the known categories, we associate each class $k \in \mathcal{Y}_s$ with a fixed *ideal prototype* $\mathbf{p}_k \in \partial \mathbb{D}_c^d$ [23], placed uniformly on the boundary of the Poincaré ball. These prototypes serve as directional anchors and remain fixed throughout training, enabling compact clustering of seen-class features while leaving the interior volume of the ball available for unknown category discovery.

To align the features $z_i$ with their respective class prototypes, we adopt a **penalized Busemann loss**:

$$\mathcal{L}_{\text{Buse}} = \log \left( \frac{\|z_i - \mathbf{p}_{y_i}\|^2}{1 - c\|z_i\|^2} \right) + \phi \log(1 - \|z_i\|^2), \tag{4}$$

where $\mathbf{p}_{y_i}$ is the prototype corresponding to the class label of $z_i$, and $\phi$ is a regularization coefficient. The first term guides directional alignment between features and their prototypes, preserving semantic proximity in the hyperbolic geometry. The second term penalizes embeddings that approach the boundary too aggressively, thereby maintaining stability during optimization and avoiding overconfidence.

### 3.3.5 Hybrid Hyperbolic Contrastive Loss

While the Busemann loss anchors known classes via directional alignment, it does not explicitly enforce local structure among unlabeled or novel samples. To address this, we incorporate a **hybrid hyperbolic contrastive loss** [22], designed to refine the latent space by encouraging consistency between augmented views and separating unrelated instances—even in the absence of explicit labels.

For each positive pair of embeddings $z_i', z_i''$, corresponding to different augmentations of the same input, we define the contrastive objective as:

$$\mathcal{L}_u = \frac{1}{|B|} \sum_{i \in B} -\log \frac{\exp(\delta(z_i'', z_i')/\tau)}{\sum_{j \neq i} \exp(\delta(z_i', z_j)/\tau)}, \tag{5}$$

where $\tau$ is a temperature hyperparameter and $B$ is the batch of samples. We use a **hybrid similarity function** $\delta(.,.)$, which linearly combines distance-based and angle-based measures:

$$\delta(.,.) = \alpha_d \cdot \underbrace{[-\mathbb{D}_\mathbb{H}(.,.)]}_{L_u^{dis}} + (1 - \alpha_d) \cdot \underbrace{\cos(.,.)}_{L_u^{ang}}, \tag{6}$$

$\cos(\cdot, \cdot)$ computes cosine similarity in the tangent space, thanks to the co-conformality of the Euclidean and Hyperbolic spaces. $\alpha_d$ is the balancing factor (for more details see **Sup. Mat.**).

This hybrid formulation leverages the metric structure of hyperbolic space to promote global semantic separation via geodesic distances, while retaining angular consistency within local neighborhoods. Fig. 3 shows the importance of the full $\delta$ over the individual distance metrics.

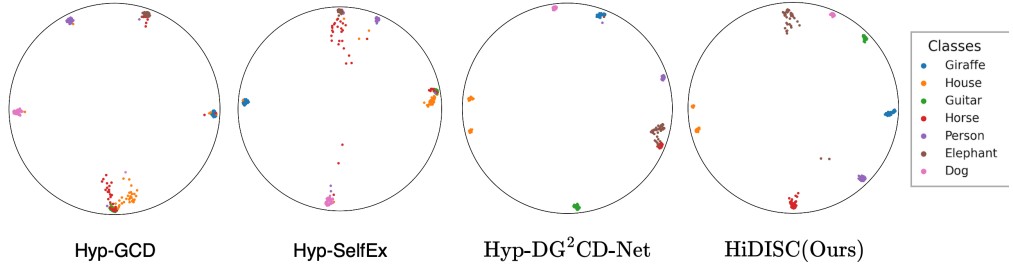

Figure 4: **Poincaré-disk UMAP [62] embeddings** of the target domain ("Photo") clusters, as produced by Hyp-GCD [22], Hyp-SelfEx [22], Hyp-DG$^2$CD-Net, and HIDISC(Ours) for the PACS dataset, with "Sketch" as the source. HIDISC produces a visually clean and compact embedding space, supported by **silhouette scores** [63] ($\in [-1.1], \uparrow$), indicating improved cluster compactness and separation: (Hyp-GCD: **-0.52**, Hyp-SelfEx: **-0.42**, Hyp-DG$^2$CD-Net: **-0.29**, HIDISC: **-0.14**)

### 3.3.6 Training Objective

Our final *minimization* objective integrates all components:

$$\mathcal{L}_{\text{total}} = \lambda_1 \underbrace{\mathcal{L}_{\text{Buse}}}_{\text{Semantic alignment}} + \lambda_2 \underbrace{\mathcal{L}_u}_{\text{Contrastive regularization}} + \lambda_3 \underbrace{\mathcal{L}_{\text{out}}}_{\text{Outlier repulsion}}, \quad \text{where } \lambda_1 + \lambda_2 + \lambda_3 = 1. \quad (7)$$

**Test-time Protocol.** After training, we extract hyperbolic features from target-domain samples and perform clustering using K-Means as in [1, 15]. See **Sup. Mat.** for detailed algorithm.

### 3.4 Theoretical Justification: Generalization in Euclidean vs. Hyperbolic Spaces

We analyze HIDISC under the lens of generalization theory in hyperbolic space. Given $\mathcal{D}_S, \mathcal{D}_T$, and synthetic augmentations $\{\mathcal{D}_{\text{syn}}^{(l)}\}_{l=1}^{\mathcal{M}}$, the goal is to minimize the expected target risk:

$$\mathcal{L}_T(f) = \mathbb{E}_{x \sim T}[\ell(f(x))], \quad (8)$$

Extending Rademacher-based analysis to the Poincaré ball $\mathbb{D}_c^d$ [21], we obtain:

$$\mathcal{L}_T(f) \leq \mathcal{L}_{S'}(f) + \Delta_{\mathbb{H}}(S', T) + \mathcal{R}_{\mathbb{H}}(\mathcal{H}) + \epsilon, \quad (9)$$

where: $\mathcal{L}_{S'}(f)$: empirical loss over the augmented training set $S'$; $\Delta_{\mathbb{H}}(S', T)$: hyperbolic discrepancy between augmented source and target; $\mathcal{R}_{\mathbb{H}}(\mathcal{H})$: Rademacher complexity of the hypothesis class $\mathcal{H}$; $\epsilon$: a residual optimization error.

Compared to the Euclidean bound $\mathcal{L}_T^{\mathbb{E}}(f) \leq \mathcal{L}_{S'}(f) + \Delta_{\mathbb{E}}(S', T) + \mathcal{R}_{\mathbb{E}}(\mathcal{H}) + \epsilon$, the hyperbolic version benefits from the exponential volume and hierarchical structure of $\mathbb{D}_c^d$. This allows semantically distant concepts to be placed further apart with less distortion and curvature-driven compression around known classes—thereby making fewer, well-chosen augmentations sufficient to span the generalization space. As such, $\Delta_{\mathbb{H}}(S', T) < \Delta_{\mathbb{E}}(S', T)$ holds under the same augmentation budget, yielding a tighter bound (see **Sup. Mat.** for a formal proof).

Each loss in HIDISC contributes to improving specific terms: **(i)** The Busemann loss $\mathcal{L}_{\text{Buse}}$ aligns seen-class features to ideal prototypes at the boundary, stabilizing $\mathcal{L}_{S'}(f)$ via directional compactness; **(ii)** The hybrid contrastive loss $\mathcal{L}_u$ integrates angular and geodesic similarity to encourage semantically meaningful clusters and reduce model complexity $\mathcal{R}_{\mathbb{H}}(\mathcal{H})$; **(iii)** The outlier loss $\mathcal{L}_{\text{out}}$, applied on Tangent CutMix samples, helps partition the open space, reducing false positives on novel categories without explicit domain alignment; **(iv)** The curated synthetic domains $\{\mathcal{D}_{\text{syn}}^{(l)}\}$ enrich $S'$, approximating $T$'s support and reducing $\Delta_{\mathbb{H}}(S', T)$ in a geometry-consistent manner.

In **Sup. Mat.**, we show that FID-based estimates of $\Delta_{\mathbb{H}}$ yield minimal improvement over the inherent domain-independence of hyperbolic geometry. We also compare our loss terms in Euclidean and hyperbolic spaces, demonstrating that hyperbolic geometry better reduces the generalization gap.

## 4 Experimental Evaluations

**Dataset Details.** We evaluate our method on three standard DG-GCD benchmarks: PACS [2], Office-Home[3], and Domain-Net[4]. We follow the protocol of [1] for constructing known/novel class-splits and source-target domain pairs. The dataset details are provided in the **Sup. Mat**.

Table 1: **Comparison of clustering accuracy** (%) for known (Old), novel (New), and overall (All) categories across PACS, Office-Home, and DomainNet. It can be seen that HιDιSC beats other synthetic domain augmentation based baselines using significantly less number of synthetic domains (from 6/9 to 2). (**Bold** : best , underline : second best).

| Method | Venue | PACS | | | Office-Home | | | DomainNet | | | Avg. | | |
|---|---|---|---|---|---|---|---|---|---|---|---|---|---|
| | | All | Old | New | All | Old | New | All | Old | New | All | Old | New |
| ViT [60] | ICLR'21 | 41.98 | 50.91 | 33.16 | 26.17 | 29.13 | 21.62 | 25.35 | 26.48 | 22.41 | 31.17 | 35.51 | 25.73 |
| GCD [15] | CVPR'22 | 52.28 | 62.20 | 38.39 | 52.71 | 54.19 | 50.29 | 27.41 | 27.88 | 26.13 | 44.13 | 48.09 | 38.27 |
| SimGCD [25] | ICCV'23 | 34.55 | 38.64 | 30.51 | 36.32 | 49.48 | 13.55 | 2.84 | 2.16 | 3.75 | 24.57 | 30.09 | 15.94 |
| CMS [26] | CVPR'24 | 28.95 | 28.13 | 36.80 | 10.02 | 9.66 | 10.53 | 2.33 | 2.40 | 2.17 | 13.77 | 13.40 | 16.50 |
| SelfEx [64] | ECCV'24 | 71.82 | 73.37 | 71.55 | 50.18 | 48.59 | 52.16 | 24.78 | 24.99 | 24.21 | 48.93 | 48.98 | 49.31 |
| CDAD-Net [18] | CVPR-W'24 | 69.15 | 69.40 | 68.83 | 53.69 | 57.07 | 47.32 | 24.12 | 23.99 | 24.35 | 48.99 | 50.15 | 46.83 |
| GCD+ 6 Synth | CVPR'22 | 65.33 | 67.10 | 64.42 | 50.50 | 51.48 | 48.96 | 24.71 | 24.80 | 21.94 | 46.85 | 47.78 | 45.11 |
| SimGCD+ 6 Synth | ICCV'23 | 39.76 | 43.76 | 35.97 | 35.57 | 48.58 | 12.89 | 2.71 | 1.99 | 4.14 | 26.01 | 31.44 | 17.67 |
| CMS+ 6 Synth | CVPR'24 | 28.01 | 26.71 | 29.04 | 12.09 | 12.66 | 11.13 | 3.22 | 3.28 | 3.03 | 14.44 | 14.22 | 14.40 |
| CDAD+ 6 Synth | CVPR-W'24 | 60.76 | 61.67 | 59.49 | 53.49 | 56.90 | 47.76 | 23.85 | 23.88 | 24.26 | 46.03 | 47.47 | 43.84 |
| Hyp-GCD [22] | CVPR'25 | 65.33 | 67.11 | 64.42 | 50.13 | 49.36 | 48.08 | 22.88 | 23.74 | 25.89 | 46.12 | 46.74 | 46.13 |
| Hyp-SelfEx [64] | ECCV'24 | 72.44 | 74.70 | 71.20 | 52.91 | 52.65 | 52.96 | 29.30 | 30.45 | 26.37 | 51.55 | 52.60 | 50.18 |
| **DG$^2$CD-Net** [1] (9 Synth) | CVPR'25 | 73.30 | 75.28 | 72.56 | 53.86 | 53.37 | **54.33** | 29.01 | 30.38 | 25.46 | 52.06 | 53.01 | 50.78 |
| Hyp-DG$^2$CD-Net$^\dagger$(9 Synth) | CVPR'25 | 74.07 | 74.40 | 73.95 | 49.40 | 50.29 | 48.03 | 22.31 | 21.52 | 24.29 | 48.59 | 48.74 | 48.76 |
| **HιDιSC (Ours)** (2 Synth) | – | **75.07** | **75.54** | **74.52** | **56.78** | **59.23** | 53.21 | **30.51** | **31.40** | **28.41** | **54.12** | **55.39** | **52.05** |
| Δ | – | +1.00 | +0.26 | +0.57 | +2.92 | +2.16 | –1.12 | +1.21 | +0.95 | +2.04 | +2.06 | +2.38 | +1.27 |
| CDAD-Net (DA) [UB] | CVPR-W'24 | 83.25 | 87.58 | 77.35 | 67.55 | 72.42 | 63.44 | 70.28 | 76.46 | 65.19 | 73.69 | 78.82 | 68.66 |

**Evaluation Metrics.** Following [15, 18, 1], we evaluate clustering using three metrics: **Old** (accuracy on known classes $\mathcal{Y}_t^{\text{old}}$), **New** (accuracy on novel classes $\mathcal{Y}_t^{\text{new}}$), and **All** (overall accuracy on $\mathcal{D}_T$). Hungarian matching is used to align predicted clusters with ground-truth labels. Scores are averaged over three runs and all source-target combinations. Further experimental details and hyper-parameter choices are mentioned in **Sup. Mat.**

### 4.1 Comparisons to the Literature

Table 1 compares our proposed HιDISC against state-of-the-art methods on the said datasets. Baselines are categorized into four groups: (i) **Euclidean source-only GCD methods**, including GCD [15], SimGCD [25], CMS [26], and SelfEx [64]; (ii) **Synthetic augmentation-based GCD methods**, such as SimGCD+Synthetic, CMS+Synthetic, and CDAD-Net+Synthetic [18], which incorporate domain-shifted images via diffusion-based generation; (iii) **Hyperbolic GCD methods**, including Hyp-GCD [22] and Hyp-SelfEx, which project features into hyperbolic space to improve clustering but do not generalize across domains. To ensure consistency with the DG-GCD setting, we retain only the components of these methods that rely on labeled data during training and omit terms involving unlabeled samples in all the above baselines, as recommended in [1]; and (iv) the **DG-GCD baseline** DG$^2$CD-Net [1], which simulates multiple domains using diffusion models and aggregates task-level knowledge via episodic training and task vectors. For a fairer comparison, we also implement a hyperbolic variant, Hyp-DG$^2$CD-Net, by replacing its embedding space with a Poincaré ball. As in [1], we report results for CDAD-Net [18] under joint access to source and target domains as an upper bound of our results.

Quantitatively, HιDISC achieves state-of-the-art performance across all metrics and datasets. It improves upon DG$^2$CD-Net by +2.06% in average overall clustering accuracy and by +1.27% on novel class discovery. On DomainNet—the most diverse and challenging benchmark—HιDISC outperforms the best previous method by +1.21%. UMAP visualizations (Fig. 4) show HιDISC forms a compact embedding space. These gains are achieved without target access and with over **96× lower training FLOPs** than [1] while using the same number of synthetic domains (see **Sup. Mat**). On the other hand, the performance of DG$^2$CD-Net degrades drastically as the number of synthetic domains is reduced (see **Sup. Mat.**)

Table 2: **Estimated number of clusters**. Correct estimates are in green, small errors in orange, and large deviations in red.

| Method | PACS | Office-Home | DomainNet |
|---|---|---|---|
| Ground Truth | 7 | 65 | 345 |
| **DG$^2$CD-Net** | 7 | 67 | 355 |
| CDAD-Net (DG) | 12 | 60 | 362 |
| CDAD-Net (DA) | 7 | 66 | 349 |
| **HιDISC (Ours)** | 7 | 66 | 351 |

Table 3: Impact of **loss components** of HɪDISC on Office-Home

| Config. | $\mathcal{L}_{\text{Buse}}$ | $\mathcal{L}_{\text{hrep}}^u$ | $\mathcal{L}_{\text{out}}$ | Office-Home | | |
|---|---|---|---|---|---|---|
| | | | | **All** | **Old** | **New** |
| **Vanilla** | ✗ | ✗ | ✗ | 26.17 | 29.13 | 21.62 |
| $\mathcal{L}_{\text{Buse}}$ | ✓ | ✗ | ✗ | 56.32 | 59.74 | 50.32 |
| $\mathcal{L}_{\text{hrep}}^u$ | ✗ | ✓ | ✗ | 50.95 | 49.33 | 53.06 |
| $\mathcal{L}_{\text{Buse}}+\mathcal{L}_{\text{hrep}}^u$ | ✓ | ✓ | ✗ | 56.29 | 60.36 | 50.41 |
| $\mathcal{L}_{\text{Buse}}+\mathcal{L}_{\text{out}}$ | ✓ | ✗ | ✓ | 51.04 | 51.51 | 50.29 |
| **Full HɪDISC** | ✓ | ✓ | ✓ | **56.78** | **59.23** | **53.21** |

Table 4: Performance metrics demonstrating the **influence of key model components** of HɪDISC for Office-Home.

| Model Variant | Office-Home | | |
|---|---|---|---|
| | **All** | **Old** | **New** |
| - With manual augmentations based $\mathcal{D}_{\text{syn}}$ | 50.80 | 51.75 | 49.15 |
| - Without synthetic domain | 56.07 | 59.29 | 50.67 |
| - Fixed curvature (c=0.01, close to Euclidean) | 56.23 | 58.65 | 52.68 |
| - Fixed curvature (c=0.03) | 55.67 | 57.39 | 52.69 |
| - Cut-Mix (In Euclidean Space) | 53.46 | 54.86 | 51.06 |
| - Full HɪDISC | **56.78** | **59.23** | **53.21** |

Furthermore, Table 2 compares the **estimated number of clusters** inferred by each method against the ground truth on PACS, Office-Home, and DomainNet. HɪDISC is found to approximate the cluster counts more precisely than the counterparts.

The **learnable curvature** converges to dataset-specific values: 0.041 for Office-Home, 0.059 for PACS, and 0.38 for DomainNet. The curvature evolution plots are mentioned in **Sup. Mat.**

### 4.2 Ablation Analysis

**Impact of Loss Components and Key Model Components.** Table 3 evaluates the contribution of each loss component in HɪDISC on Office-Home. The vanilla model, trained without any loss terms, yields only 26.17% overall accuracy. Introducing the Busemann loss alone improves performance substantially to 56.32%, while the hybrid hyperbolic contrastive loss independently achieves 50.95%. Combining both leads to further gains, particularly in old-class accuracy. Incorporating the outlier repulsion term yields the best overall result, with 56.78% total accuracy, 59.23% on known classes, and 53.21% on novel classes.

Table 4 presents ablations on key architectural components. Manual augmentations achieve 50.80% accuracy. Replacing the learnable curvature with static values ($c = 0.01$ and $0.03$) reduces accuracy considerably, as it fails to manage the data manifold effectively. Substituting Tangent CutMix with Euclidean CutMix lowers performance by 3.96%, confirming the benefits of curvature-consistent mixing. The complete HɪDISC configuration consistently outperforms all variants, confirming the complementary benefits of its geometric and loss-driven design.

**Ablation of Norm Radius and Slope in Hyperbolic Embedding.** We study two key hyperparameters in our hyperbolic embedding setup: the $\ell_2$ norm radius before exponential mapping and the slope $\phi$ in the penalized Busemann loss, both controlling embedding compactness and placement. As per Table 5, lower slopes (e.g., $\phi = 0.10$) favor seen-class accuracy but hurt generalization, while higher slopes (e.g., $\phi = 0.90$) improve novel-class performance by restricting dispersion. We choose $\phi = 0.75$ for balance. For the norm radius, Table 5 shows that $1.5$ best balances alignment to boundary-anchored prototypes and generalization, whereas smaller values (e.g., $1.0$) overfit $\mathcal{Y}_S$.

Table 5: **Ablation on hyperbolic embedding parameters on Office-Home.** (**Left**) Effect of slope coefficient $\phi$ in the penalized Busemann loss. Lower $\phi$ concentrates embeddings near the boundary, improving seen-class accuracy but reducing generalization. (**Right**) Effect of $\ell_2$ radius constraint before exponential mapping. Radius = 1.5 yields the best trade-off between known and novel categories.

| Slope $\phi$ | All | Old | New |
|---|---|---|---|
| 0.10 | 58.84 | 65.77 | 47.07 |
| 0.75 | **56.78** | **59.23** | **53.21** |
| 0.90 | 57.76 | 62.82 | 49.18 |

| Radius | All | Old | New |
|---|---|---|---|
| 1.5 | **56.78** | **59.23** | **53.21** |
| 1.0 | 57.33 | 61.14 | 51.76 |
| 2.3 | 57.31 | 60.96 | 52.04 |

**Choice of Hyperbolic Model: Poincaré vs. Lorentz Model**

We adopt the Poincaré ball model, which we find performs more favorably than the Lorentz model [65] for DG-GCD on Office-Home in Table 6. Full theoretical details are in the **Sup. Mat.** The empirical comparison is below:

**Hyperparameter Sensitivity for Loss Weights.** We conducted an ablation study on the loss weights $(\lambda_1, \lambda_2, \lambda_3)$ and found our chosen configuration achieves near-optimal performance, demonstrating robustness Table 7. More details are in the **Sup. Mat.**

Table 6: Comparison of Poincaré ball and Lorentz models on Office-Home.

| Model | All | Old | New |
|---|---|---|---|
| Poincaré Ball [20] | **56.78** | **59.23** | **53.21** |
| Lorentz Model [65] | 54.28 | 56.01 | 51.41 |

Table 7: Ablation study on loss term weights $(\lambda_1, \lambda_2, \lambda_3)$ on OfficeHome.

| | Loss Weights | | | Acc. (%) | | |
|---|---|---|---|---|---|---|
| Config. | $\lambda_1$ | $\lambda_2$ | $\lambda_3$ | All | Old | New |
| **Config. 1** | 0.60 | 0.25 | 0.15 | **56.78** | **59.23** | **53.21** |
| **Config. 2** | 0.15 | 0.60 | 0.25 | 52.12 | 53.33 | 50.07 |
| **Config. 3** | 0.25 | 0.15 | 0.60 | 51.37 | 52.17 | 50.01 |

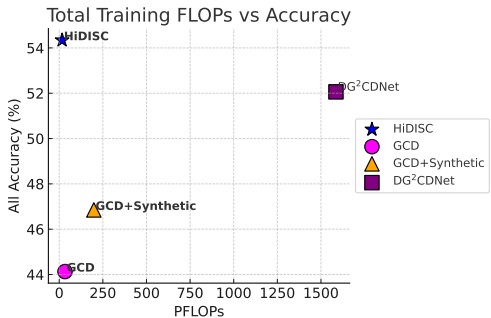

Figure 5: **Computational efficiency of HIDISC.** Hyp-Busemann requires only **16.53 PFLOPs** over 50 epochs with a batch size of $128 \times 2$, representing a $\sim 2\times$ reduction compared to GCD (33.06 PFLOPs), $\sim 12\times$ vs. GCD+Synthetic (198.36 PFLOPs), and nearly $\sim 96\times$ vs. DG$^2$CD-Net (1,586 PFLOPs). Despite this efficiency, Hyp-Busemann maintains superior accuracy without relying on episodic-training loops, simplifying the overall training pipeline.

## 5 Takeaways

We addressed the problem of DG-GCD, where novel categories emerge in unseen domains without target supervision. To this end, we proposed HIDISC, a hyperbolic representation learning framework that leverages penalized Busemann alignment, Tangent CutMix-based augmentation, and hybrid contrastive regularization to enable domain- and category-level generalization. Extensive experiments across PACS, Office-Home, and DomainNet show that HIDISC achieves state-of-the-art performance, particularly improving novel class discovery under domain shift. Our findings underscore the utility of hyperbolic geometry for scalable open-world recognition. **Future directions** include extending HIDISC to continual DG-GCD and integrating it with large-scale vision-language models.

**Broader Impact and Limitations:** While HIDISC advances open-world recognition under domain shift using geometry-aware learning, which is extremely practical, its reliance on synthetic augmentations guided by diffusion models may limit applicability in resource-constrained or safety-critical environments where generative artifacts could propagate bias.

## Acknowledgments and Disclosure of Funding

We thank our colleague Shubranil B. for his assistance with the figures in this paper. We are also grateful to Adobe Research and the CMInDS department for providing the necessary resources and support.

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
