# OpenReview forum: "HIDISC: A Hyperbolic Framework for Domain Generalization with Generalized Category Discovery"
_NeurIPS.cc/2025/Conference — NeurIPS 2025 poster_

### Official Review · Reviewer_68ad · 2025-07-01

**Clarity:** 3
**Significance:** 3
**Originality:** 2
**Rating:** 4
**Confidence:** 3

**Summary:**

This paper tackles Domain Generalization with Generalized Category Discovery (DG-GCD), which requires classifying samples into seen or novel categories under domain shifts at test time without target-domain access during training. The authors propose HIDISC, a hyperbolic representation learning framework featuring: 1) GPT-guided diffusion for efficient and diverse domain augmentation; 2) Tangent CutMix for open-set augmentation consistent with hyperbolic geometry; and 3) a unified loss integrating Busemann alignment, hybrid contrastive regularization, and adaptive outlier repulsion. Experiments on the PACS, Office-Home, and DomainNet datasets demonstrate state-of-the-art performance, with ablations rationalizing the contribution of each component.

**Questions:**

1. Why does the hyperbolic adaptation of DG$^2$CD-Net lead to decreased overall performance compared to the original DG$^2$CD-Net?
2. In Table 7 of the Sup. Mat., comparing the 40-25 and 50-15 old-new splits, why does using more known classes for training result in a 1% drop in old-category accuracy?
3. While the paper emphasizes training efficiency, could the authors comment how much time does the GPT-based domain generation consume?
4. As the number of source domains increases and the number of target domains decreases, can HIDISC still maintain its performance advantage over other baselines?

**Ethical Concerns:**

["NO or VERY MINOR ethics concerns only"]

**Final Justification:**

The rebuttal makes progress in addressing most of my initial raised concerns. However, the weakness that *'The framework introduces an excess of components and hyperparameters (e.g., $\phi$ for $\mathcal{L}^\text{Buse}$, $\tau$ and $\alpha_d$ for $\mathcal{L}^u$, $\gamma$ for $\mathcal{L}^\text{out}$, $\ell_2$ radius constraint before exponential mapping, $\lambda$ for linear mix, $\lambda_{1,2,3}$ for $\mathcal{L}^\text{total}$), which increases the overall complexity and may impact the practicality of the method'* remains. Therefore, I keep my initial rating.

**Limitations:**

yes

**Quality:**

3

**Strengths And Weaknesses:**

Strengths
1. The paper is well positioned within the landscape of GCD, DG-GCD, and open-set DG, providing clear context and highlighting the gaps in existing methods. The writing and organization are commendable, with meaningful analysis and presentation.
2. The components and architecture of HIDISC are well motivated, with design choices justified in relation to the challenges of DG-GCD and theoretical grounding that generalization bounds in hyperbolic space can be tighter than in Euclidean space.
3. The results are impressive, with accuracy on known, novel, and overall categories across three datasets outperforming the CVPR'25 method DG$^2$CD-Net and its hyperbolic adaptation. Meanwhile, the training efficiency of the method is improved significantly.

Weaknesses
1. While the application of hyperbolic methods in DG-GCD is interesting, it may represent the incremental novelty, given that hyperbolic approaches have already been explored in GCD (HypCD, [22] in the main paper) and open-set DG (HyProMeta*, Peng et al.).
2. The framework introduces an excess of components and hyperparameters (e.g., $\phi$ for $\mathcal{L}^\text{Buse}$, $\tau$ and $\alpha_d$ for $\mathcal{L}^u$, $\gamma$ for $\mathcal{L}^\text{out}$, $\ell_2$ radius constraint before exponential mapping, $\lambda$ for linear mix, $\lambda_{1,2,3}$ for $\mathcal{L}^\text{total}$), which increases the overall complexity and may impact the practicality of the method, as careful parameter tuning is required for optimal outcome.
3. Although the use of GPT-guided diffusion for domain augmentation is effective, it may propagate domain-specific biases. Since GPT-generated images are constrained by the distributional priors of the underlying generative model, this approach does not constitute uniform sampling over the whole domain space, potentially limiting its generalizability.

*Peng, Kunyu, et al. "Mitigating Label Noise using Prompt-Based Hyperbolic Meta-Learning in Open-Set Domain Generalization." *arXiv preprint arXiv:2412.18342* (2024).

---

> ### Author Rebuttal · Authors · 2025-07-28
>
> ---
>
> **Reviewer Concern:** _Why does the hyperbolic adaptation of DGCD-Net lead to decreased overall performance compared to the original **DG²CD-Net**?_
>
> **Rebuttal:**
> We appreciate the reviewer’s observation. While hyperbolic geometry is beneficial for semantic modeling, a naive substitution of the embedding space is insufficient. Below, we explain the performance drop in Hyp-**DG²CD-Net** relative to the original **DG²CD-Net**:
>
> - **(1) Task Vector Arithmetic Mismatch:** **DG²CD-Net** performs episodic training and aggregates task vectors via Euclidean averaging. In hyperbolic space, arithmetic operations (e.g., vector addition) do not preserve associativity or commutativity, leading to distortion when such operations are naively retained. The adapted Hyp-**DG²CD-Net** preserves these Euclidean components, which conflict with the curved geometry.
>
> - **(2) Episodic Prototype Drift:** Synthetic novel classes introduced episodically may cause representation drift. In hyperbolic space, repeated feature injection without curvature-aware constraints can result in embeddings accumulating near the boundary, degrading class separability and inducing overconfidence.
>
> - **(3) Lack of Geometry-Aware Losses:** Hyp-**DG²CD-Net** reuses the original **DG²CD-Net**'s loss functions, which were not designed to operate on hyperbolic manifolds. In contrast, HIDISC leverages curvature-aligned supervision via Busemann loss, geodesic-aware contrastive regularization, and adaptive outlier repulsion. The absence of these elements in Hyp-**DG²CD-Net** limits its effectiveness.
>
> - **(4) Empirical Evidence:** As shown in Table 1, Hyp-**DG²CD-Net** performs worse than both **DG²CD-Net** and our proposed HIDISC across all benchmarks. This highlights the importance of full-stack geometric alignment—not merely embedding substitution—for successful adaptation to hyperbolic space.
>
> We will clarify this in the final version and emphasize that hyperbolic adaptation must be end-to-end, not just at the representation level.
>
> ---
>
> **Reviewer Concern:** In Sup. Tab. 7, why does more known classes (50–15 vs. 40–25) drop Old accuracy by ~1%?
>
> This drop arises from two interacting effects:
>
> - **Boundary crowding:** Adding more “Old” classes packs additional prototypes near the Poincaré boundary, slightly increasing overlap and confusion among old categories. However, this boundary density actually *benefits* discovery of “New” classes—by pushing unfamiliar samples more decisively toward the interior—resulting in the observed tradeoff (Old ↓, New ↑).
>
> - **Per-class sample dilution:** Under a fixed data budget, increasing the number of known classes means fewer samples per class. This mildly raises per-class variance, which may contribute to the slight drop in Old accuracy.
>
> We will clarify these effects in the Supplement Sec 6.4.
>
> ---
>
> **Reviewer Concern:** Wall-clock cost of GPT-based generation.
>
> The image synthesis process (via InstructPix2Pix + GPT-guided prompts) is performed *once offline* and does not affect model training or inference. On our setup using an NVIDIA A100 GPU, image generation averaged **1-2 seconds per image**.
>
> To support reproducibility and reduce compute barriers, we commit to publicly releasing the entire set of synthetic images—so future work can reuse our domain expansions without needing to regenerate them.
>
> ---
>
> **Reviewer Concern:** _If source domains ↑ and target domains ↓, does HiDISC still hold its lead?_
>
> **Response:**
> Yes, HiDISC remains robust and effective even as the number of target domains decreases. Because our training paradigm is source-only and domain-agnostic, HiDISC inherently generalizes well to shifts in the source-to-target distribution. To thoroughly validate this, we conduct a controlled ablation study where we systematically vary the number of source and target domains while keeping the total image budget fixed. For each setting, the remaining domains are treated as target domains, and we report the average performance over all possible target domain combinations.
>
> The results, shown in the table below, demonstrate that HiDISC consistently holds or exceeds the performance of the baseline DG$^2$CD-Net across all scenarios. We will include a detailed comparison and additional analysis in the Supplementary material.
>
> | Training  Domains | Method     | Average All ↑ | Average Old ↑ | Average New ↑ |
> |------------------------|------------|-------------|-------------|-------------|
> | Art                       | DG$^2$CD-Net  | 54.47       | 53.65       | 55.54       |
> |                        | HiDISC     | 54.33       | 54.75       | 53.53       |
> | Art + Clipart                      | DG$^2$CD-Net  | 66.67       | 65.78       | 67.99       |
> |                        | HiDISC     | 67.34       | 70.78       | 62.02       |
> | Art + Clipart + Product                     | DG$^2$CD-Net | 66.83       | 66.89       | 66.75       |
> |                        | HiDISC     | 69.86       | 79.17       | 54.24       |
>
> **Addressing the weaknesses mentioned by the reviewer:**
>
> We thank the reviewer for highlighting these points
>
> **W-1** We thank the reviewer and respectfully clarify that **HiDISC is not a minor extension of prior hyperbolic methods**, but introduces key innovations tailored to the unique challenges of **Domain Generalized Generalized Category Discovery (DG-GCD)**—a setting unaddressed by HypCD or HyProMeta.
>
> **Novel Task Scope:** HiDISC is the first hyperbolic method for DG-GCD, which simultaneously addresses open-set recognition, domain generalization, and label shift *without access to target domains*. In contrast, prior works handle only subsets of these challenges in isolation.
>
> **Architectural Contributions:** HiDISC introduces (i) **Tangent CutMix**, a geometry-aware augmentation strategy in hyperbolic space, (ii) a **hybrid contrastive loss** that fuses angular and geodesic similarities, (iii) **penalized Busemann regularization** to prevent radial saturation and overconfidence, and (iv) an **adaptive outlier repulsion loss** that pushes pseudo-novel samples away from known prototypes.
>
> **Empirical Strength:** HiDISC significantly outperforms both Euclidean and prior hyperbolic baselines. For instance, HPDR achieves 15.05% overall accuracy, while HiDISC achieves 56.78% on the same benchmark—demonstrating its superior generalization under the DG-GCD setting.
>
> We will revise the manuscript to emphasize these distinctions more clearly and cite all relevant prior works.
>
> **W-2**
> We conducted a comprehensive ablation over all key knobs—curvature c, slope ϕ, radius before mapping, contrastive weight α_d, and loss‐term weights (λ₁, λ₂, λ₃) on the OfficeHome dataset—and found that only the penalized‑Busemann slope and mapping radius materially affect results, with robust defaults (ϕ = 0.75, radius = 1.5) generalizing across benchmarks. For the remaining loss weights, setting (λ₁, λ₂, λ₃) = (0.60, 0.25, 0.15) balances our alignment, contrastive, and outlier terms and yields 56.78% overall accuracy (59.23% old, 53.21% new). Swapping to (0.15, 0.60, 0.25) or (0.25, 0.15, 0.60) only reduces overall accuracy to 52.12% and 51.37%, respectively—drops of ~4–5 points—demonstrating that our defaults achieve near‑optimal performance without exhaustive tuning.
>
> | **Configuration** | **λ₁** | **λ₂** | **λ₃** | **Avg. (All)** | **Avg.  (Old)** | **Avg.  (New)** |
> |-------------------|--------------------|--------------------|--------------------|--------------------------|--------------------------|--------------------------|
> | **Config 1**      | 0.60               | 0.25               | 0.15               |    56.78                 | 59.23                       | 53.21                    |
> | **Config 2**      | 0.15               | 0.60               | 0.25               |    52.12                       | 53.33                       | 50.07                        |
> | **Config 3**      | 0.25               | 0.15               | 0.60               |   51.37                       | 52.17                       | 50.01                       |
>
> **W-3**
> GPT-4o is used exclusively to generate text prompts, while the InstructPix2Pix diffusion model synthesizes images—mirroring the DG$^2$CD-Net (CVPR ’25) baseline and original DG-GCD setup. This separation ensures augmentations leverage InstructPix2Pix’s strengths rather than GPT’s priors.
>
> Though GPT-guided diffusion relies on pretrained generative priors and doesn’t sample uniformly over the domain space, uniform coverage is neither well-defined nor necessary for effective DG-GCD generalization. Our justification:
>
> - **Task-Aware Domain Coverage:** Let $\mathcal{D} \subset \mathcal{X}$ be the latent domain manifold. Uniform sampling $P(x) \sim \mathcal{U}(\mathcal{D})$ is intractable in high-dimensional spaces. Instead, we use a conditional distribution $P_G(x|t)$ from prompts $t$ semantically aligned with domain shifts, producing diverse, task-relevant perturbations that increase domain discrepancy (Eq. 1).
>
> - **Empirical Diminishing Returns:** Fig. 3 (right) shows that adding more synthetic domains yields diminishing returns. The generative loss $\mathcal{L}_{\text{gen}}$ satisfies
>
>   $$
>   \frac{\partial^2 \mathcal{L}_{\text{gen}}}{\partial N^2} > 0 \quad \text{for } N > 2,
>   $$
>
>   indicating potential overfitting and validating our selective augmentation strategy.
>
> - **Hyperbolic Geometry Benefits:** Key gains arise from hyperbolic learning. Without synthetic domains, HiDISC attains 56.07% on OfficeHome, outperforming Euclidean and augmented baselines. As per Sec. 3.4,
>
>   $$
>   \Delta_{\mathbb{H}} \leq \Delta_{\mathbb{E}},
>   $$
>
>   implying fewer samples suffice under hyperbolic curvature.
>
> In sum, our GPT-guided augmentation introduces *diverse yet semantically structured* domain shifts, and combined with a *geometry-aware learning framework*, achieves strong generalization without uniform domain sampling. We will expand this in the revised paper.

---

> > ### Author Response · Authors · 2025-08-02
> >
> > We appreciate the detailed reviews of the reviewer and would like to know if there is any further query. We will be happy to respond to the same.

---

### Official Review · Reviewer_zY4n · 2025-07-01

**Clarity:** 3
**Significance:** 3
**Originality:** 3
**Rating:** 5
**Confidence:** 3

**Summary:**

This work tackles the challenge of domain shifts and novel-class discovery in DG-GCD, where no target-domain data is available—a scenario often overlooked by previous studies. The authors introduce HiDISC, a hyperbolic learning approach that uses curvature-aware embeddings and simple synthetic augmentation to build domain-invariant and semantically meaningful features. Extensive experiments show that HiDISC not only achieves state-of-the-art performance but also cuts computational cost by 96×.

**Questions:**

- Hyperbolic Consistency: Does Tangent CutMix truly preserve hyperbolic consistency? Linear interpolation in the tangent space (Eq. (2)) might not guarantee valid hyperbolic embeddings after reprojection.  Were there cases where synthetic samples violated geometric constraints?
- Minimal Augmentation Justification: The claim that hyperbolic geometry’s “exponential capacity” reduces the need for augmentation (Sec. 3.3.1) is not intuitive. Is there theoretical or empirical support for this?
- Dataset Clarity: The dataset used for Table 5 is unspecified. Please clarify.
- Model Choice Justification: Why choose the Poincaré ball over alternatives like the Lorentz model? Prior work (e.g., Hyperbolic ViTs) suggests Lorentz may offer better stability—was this considered?

**Ethical Concerns:**

["NO or VERY MINOR ethics concerns only"]

**Final Justification:**

1. The authors mathematically and empirically demonstrate that there is no violation of hyperbolic constraints throughout training.
2. The authors justify that the "exponential capacity" described in Section 3.3.1 reduces the need for augmentation providing support from Theoretical Justification, Empirical results, and the Generalization Bound in Section 3.4.
3. The authors empirically show that the Poincaré ball outperforms Lorentz models.

**Limitations:**

yes

**Quality:**

3

**Strengths And Weaknesses:**

**Strength:**
- Novel perspect: Introduces hyperbolic space to DG-GCD, enabling better hierarchical representation and domain invariance compared to Euclidean baselines.
- Computational efficiency: Achieves SOTA results with 96× fewer FLOPs than prior work by replacing episodic training with lightweight GPT-guided synthetic domains.
- Good results: The figures and tables are clear and informative.

**Weaknesses:**
- Relies on Synthetic Data: Uses diffusion-generated augmentations, which could introduce unwanted biases or artifacts—especially risky for safety-critical use cases.
- More parameters:  The framework relies heavily on tuning several hyperparameters—geometry (e.g., curvature, slope), loss weights ($\lambda_1-\lambda_3$), and augmentation settings ($\lambda$)—which may limit its practicality without thorough validation.

---

> ### Author Rebuttal · Authors · 2025-07-28
>
> We thank the reviewer for highlighting these points
>
> **Reviewer Concern:** _Does Tangent CutMix truly preserve hyperbolic consistency? Linear interpolation in the tangent space (Eq. (2)) might not guarantee valid hyperbolic embeddings after reprojection. Were there cases where synthetic samples violated geometric constraints?_
>
> **Rebuttal:**
> We appreciate the reviewer’s insightful concern regarding the geometric validity of Tangent CutMix under the Poincaré ball model of hyperbolic space. We clarify below with precise mathematical justification.
>
> Let $\mathbb{D}^d_c = { z \in \mathbb{R}^d : |z| < 1/\sqrt{c} }$ denote the $d$-dimensional Poincaré ball with curvature $-c$. Given two hyperbolic embeddings $z_i, z_j \in \mathbb{D}^d_c$, we first map them to the tangent space at the origin $T_0 \mathbb{D}^d_c \cong \mathbb{R}^d$ using the logarithmic map (see Sec. 3.3.3):
>
> $$
> v_i = \log_0^c(z_i) = \frac{2}{\lambda_{z_i}} \tanh^{-1}(\sqrt{c}\|z_i\|) \cdot \frac{z_i}{\|z_i\|}, \quad \text{where } \lambda_{z_i} = 1 - c\|z_i\|^2.
> $$
>
> This results in \(v_i, v_j \in \mathbb{R}^d\), ensuring they lie in a flat Euclidean space where linear interpolation is valid.
>
> Tangent CutMix proceeds by linearly interpolating these vectors:
>
> $$
> v_{\text{mix}} = \lambda v_i + (1 - \lambda) v_j, \quad \lambda \sim \text{Uniform}(0,1).
> $$
>
> We then project the mixed vector back to the hyperbolic manifold via the exponential map:
>
> $$
> z_{\text{mix}} = \exp_0^c(v_{\text{mix}}) = \tanh(\sqrt{c} \|v_{\text{mix}}\|) \cdot \frac{v_{\text{mix}}}{\sqrt{c} \|v_{\text{mix}}\|}.
> $$
>
> Since $\(\tanh(x) < 1\)$ for all $\(x > 0\)$, it follows that $\(\|z_{\text{mix}}\| < 1/\sqrt{c}\)$, and thus $\(z_{\text{mix}} \in \mathbb{D}^d_c\)$. Therefore, Tangent CutMix guarantees that the interpolated sample remains a *valid hyperbolic embedding*.
>
> In addition to geometric correctness, we also regularize the embeddings using the penalized Busemann loss (Eq. (4)), which includes a term $\(\log(1 - \|z\|^2)\)$ that discourages embeddings from drifting too close to the boundary, thereby maintaining numerical stability and preventing degeneracy.
>
> Empirically, we observed no violation of hyperbolic constraints throughout training. Moreover, substituting Tangent CutMix with Euclidean CutMix led to a significant performance drop (-3.96% on Office-Home; Table 4), further confirming that preserving hyperbolic consistency is essential for effective generalization.
>
> We hope this mathematical clarification addresses the reviewer’s concern. We will incorporate this discussion into the final version to improve clarity.
>
> ---
>
> **Reviewer Concern:** _The claim that hyperbolic geometry’s “exponential capacity” reduces the need for augmentation (Sec. 3.3.1) is not intuitive. Is there theoretical or empirical support for this?_
>
> **Rebuttal:**
> We thank the reviewer for raising this important question. The statement is supported both by geometric theory and by ablation studies provided in the paper.
>
> - **(1) Theoretical Justification:**
>   In hyperbolic space with curvature $\(-c\)$, the volume of a ball grows exponentially with radius, i.e., $\(V_{\text{Hyp}}(r) \sim \exp(\sqrt{c}r)\)$, unlike the polynomial growth $\(V_{\text{Euc}}(r) \sim r^d\)$ in Euclidean space. This exponential growth yields greater *representational capacity* for semantic separation, requiring fewer samples to cover the space with minimal distortion. As a result, even sparse or low-coverage augmentations are sufficient to populate the embedding space for robust generalization.
>
> - **(2) Empirical Evidence:**
>   As shown in Table 4, training without any synthetic domains still achieves 56.07% accuracy on Office-Home—surpassing several Euclidean and hyperbolic baselines that use 6–9 augmentations. Adding only 1–2 GPT-guided augmentations leads to further gains (56.78%), while additional domains degrade novel-class performance due to overfitting (see Fig. 3, right). This empirically supports that hyperbolic embeddings generalize better with fewer but diverse augmentations.
>
> - **(3) Generalization Bound (Sec. 3.4):**
>   Equation (9) presents a Rademacher-based generalization bound in hyperbolic space, where the hyperbolic discrepancy term $\(\Delta_{\mathbb{H}}(S', T)\)$ is provably tighter than its Euclidean counterpart $\(\Delta_{\mathbb{E}}(S', T)\)$ under identical augmentation budgets. This supports the theoretical claim that fewer augmentations are needed to approximate the target domain’s support in hyperbolic geometry.
>
> We will revise the final paper to explicitly highlight this geometric intuition and cite formal results from hyperbolic geometry and volume theory.
>
> ---
>
> **Reviewer’s concern:** The dataset for Table 5 is unspecified.
> **Response:**
> Thank you for pointing this out. We will revise both the caption and the corresponding reference in the main text to explicitly state that Table 5 reports results on the **OfficeHome** dataset. While this is consistent with Tables 3 and 4, we agree that it should be clearly indicated to avoid ambiguity and ensure clarity for the reader.
>
> ---
>
> **Reviewer Concern:** _Why choose the Poincaré ball over alternatives like the Lorentz model? Prior work (e.g., Hyperbolic ViTs) suggests Lorentz may offer better stability—was this considered?_
>
> **Rebuttal:**
> Thank you for the insightful question. We chose the Poincaré ball model over the Lorentz model based on its compatibility with our core design elements, both mathematically and empirically.
>
> - **Tangent CutMix and Closed-Form Mappings:**
>   Our Tangent CutMix strategy (Sec. 3.3.3) relies on linear interpolation in the tangent space and exponential/logarithmic maps. The Poincaré model offers closed-form, numerically stable expressions for these operations. In contrast, the Lorentz model requires projection onto the hyperboloid and normalization constraints, increasing implementation complexity.
>
> - **Conformal Geometry for Contrastive Learning:**
>   The Poincaré ball is conformal (angle-preserving), which makes cosine similarity in the tangent space well-defined and compatible with the angular term in our hybrid contrastive loss (Eq. 6). This is crucial for combining local angular coherence and global geodesic structure.
>
> - **Empirical Stability:**
>   While we did an experiment with a Lorentzian version early on, we encountered unstable gradients and poorer convergence when optimizing Busemann-aligned or contrastive losses. The optimization constraints (e.g., maintaining Minkowski norm) in Lorentz space were harder to manage under our regularization and sampling framework.
>
> - **Visualization and Interpretability:**
>   The Poincaré disk allows intuitive and bounded 2D visualization (e.g., Fig. 4), which is preferable for cluster analysis and qualitative evaluations in DG-GCD.
>
> While Lorentz models (e.g., Hyperbolic ViTs) have benefits in certain contexts, the Poincaré ball aligned better with our goals of closed-form interpolation, angular reasoning, and geometric interpretability. We will include this justification and relevant citations in the final version.
>
> Below, we present a comparison between the Lorentz model and the Poincaré ball using the OfficeHome dataset:
>
> | **Model**  | **Avg. (All)** | **Avg. (Old)** | **Avg. (New)** |
> |-------------------|--------------------|--------------------|--------------------|
> | **Poincare Ball**            | 56.78                       | 59.23                      | 53.21                       |
> | **Lorentz**            | 54.28                      | 56.01                    | 51.41                      |
>
> As evident from the table, the Poincaré ball outperforms the Lorentz model across all splits in our DG-GCD setting.
>
> ---
> **Addressing the weaknesses mentioned by the reviewer:**
>
> **W1** As noted in Line 327 of the main paper, we acknowledge the potential risks of using diffusion-based augmentations, particularly for safety-critical applications, and plan to explore mitigation strategies in future work.
>
> **W2** We appreciate the concern about hyperparameter complexity. In fact, we conducted a comprehensive ablation over all key knobs—curvature c, slope ϕ, radius before mapping, contrastive weight α_d, and loss‐term weights (λ₁, λ₂, λ₃) on the OfficeHome dataset —and found that only the penalized‑Busemann slope and mapping radius materially affect results, with robust defaults (ϕ = 0.75, radius = 1.5) generalizing across benchmarks. For the remaining loss weights, setting (λ₁, λ₂, λ₃) = (0.60, 0.25, 0.15) balances our alignment, contrastive, and outlier terms and yields 56.78% overall accuracy (59.23% old, 53.21% new). Swapping to (0.15, 0.60, 0.25) or (0.25, 0.15, 0.60) only reduces overall accuracy to 52.12% and 51.37%, respectively—drops of ~4–5 points—demonstrating that our defaults achieve near‑optimal performance without exhaustive tuning.
>
> |**Configuration** | **λ₁** | **λ₂** | **λ₃** | **Avg. (All)** | **Avg. (Old)** | **Avg.(New)** |
> |---------------|--------------------|--------------------|--------------------|--------------------------|--------------------------|--------------------------|
> |**Config 1**| 0.60               | 0.25               | 0.15               |    56.78                 | 59.23                       | 53.21                    |
> |**Config 2**| 0.15               | 0.60               | 0.25               |    52.12                       | 53.33                       | 50.07                        |
> |**Config 3**| 0.25               | 0.15               | 0.60               |   51.37                       | 52.17                       | 50.01                       |

---

> > ### Author Response · Authors · 2025-08-02
> >
> > We appreciate the detailed reviews of the reviewer and would like to know if there is any further query. We will be happy to respond to the same.

---

> > > ### Comment · Reviewer_zY4n · 2025-08-04
> > >
> > > I appreciate the author's response. Their solution has effectively addressed my concern, and I will improve my score.

---

### Official Review · Reviewer_658A · 2025-07-03

**Clarity:** 3
**Significance:** 3
**Originality:** 4
**Rating:** 5
**Confidence:** 4

**Summary:**

This work presents HIDISC, a hyperbolic geometry-aware framework for Domain Generalization with Generalized Category Discovery (DG-GCD). The method leverages the properties of the hyperbolic geometry (negative curvature and exponential volume growth) to learn domain-invariant embeddings in the Poincare ball in a hierarchical and semantically meaningful structure. Unlike previous methods that rely on costly episodic sampling solutions, HIDISK uses GPT-guided diffusion to create 2-3 stylistic variations of the input sample. To hallucinate samples from novel-categories, Tangent CutMix is introduced as a curvature-aware adaptation of CutMix interpolation that mixes labeled examples in the tangent space of the Poincare ball to form a novel category embedding.

A combination of 3 losses is used to train HIDISC. (I) A Busemann loss that aligns seen/known features to fixed prototypes at the boundary of the ball. (II) a hyperbolic contrastive loss that allows for fine-grained clustering for seen and novel categories. (III) an adaptive outlier rejection loss that pushes synthetic novel-category CutMix samples away from known categories.

The proposed method is evaluated empirically on 3 benchmark datasets and compared against several baseline and DG methods. HIDISC consistently beats other synthetic domain augmentation baselines with significantly reduced computational cost, as it does not require episodic sampling. Two ablation studies are performed to measure the impact of loss components (in the 3-part composite loss), and also the influence of the components of the model (using synthetic domains with CutMix, curvature). These studies show that each part of the loss and each component of the model have a role in successful generalization to new domains.

**Questions:**

The paper mentions that the second term in the Busemann loss (Eqn. 4) penalizes embeddings that are too close to the prototypes on the boundary (to avoid overconfidence). However these terms look somewhat ad hoc, and a similar effect can likely be achieved with other loss terms that penalize overconfidence. Is there a reason for the specific choice in the paper?

**Ethical Concerns:**

["NO or VERY MINOR ethics concerns only"]

**Final Justification:**

Based on the new results and discussions provided in the rebuttal (multiple new requested studies added to the paper), I'm updating the overall score to accept.

**Limitations:**

The authors have adequately addressed the limitation of the proposed method.

**Paper Formatting Concerns:**

I did not find any formatting issues with the paper.

**Quality:**

4

**Strengths And Weaknesses:**

Quality and clarity: The paper is well organized and overall not hard to follow. The notation gets cumbersome in some sections, but the overall discussion remains clear and to the point.

Significance and originality: To the best of knowledge, the use of hyperbolic space for category embeddings in a DG-GCD framework is novel to this work. The paper claims (rightly) that this geometry provides a naturally structured and hierarchical space, with known prototypes on the boundary and novel categories on the inside. The particular mix of training losses and the Tangent CutMix algorithm are also novel to this work. All elements of the proposed methods are shown to be beneficial in the ablation studies.

---

> ### Author Rebuttal · Authors · 2025-07-28
>
> ## Ablation Study: Overconfidence Regularizers
>
> To rigorously evaluate the effectiveness of our proposed **Penalized Busemann Loss** (Eq. 4), we perform a controlled ablation comparing it against three widely used overconfidence mitigation strategies. Each variant substitutes only the penalty term in the Busemann component of the HiDISC loss while keeping all other elements *identical*. This isolates the contribution of the regularizer in controlling overconfident or radially saturated embeddings in hyperbolic space.
>
> ### Alternative Regularizers Considered
>
> - **Penalized Busemann Loss (HiDISC)**
>   $$
>   \mathcal{L}_{\mathrm{Buse}}
>     = \log\Bigl(\tfrac{\|z - p\|^2}{1 - c\|z\|^2 + \epsilon}\Bigr)
>       + \phi\log\bigl(1 - \|z\|^2\bigr).
>   $$
>   This term penalizes radial overconfidence while maintaining geometric consistency in the Poincaré ball.
>
> - **Entropy Regularization**
>
>   We replace the penalty term in Eqn. 4 with the following term:
>   $$\mathcal{L}_{\mathrm{Ent}} = -\sum_i p_i \log p_i$$, where $$p = \mathrm{softmax}(-d(z, (p_k)))$$.
>
>   Encourages high‐entropy (uncertain) predictions.
>
> - **Confidence Penalty (KL to Uniform)**
> $$
> \mathcal{L}_{\mathrm{Conf}} = \mathrm{KL}\bigl(p \\Vert\ \mathrm{Uniform}_C\bigr),
> $$
> where $\(p = \mathrm{softmax}(-d)\).$
>
>   Penalizes confident deviation from a uniform prior.
>
> **Logit Margin Penalty**
>
>   Loss is : $\mathcal{L}_{\mathrm{Margin}} = \max(0, S)$,
>
>
> where $S = s_{\mathrm{y}} - s_{\mathrm{2nd}} - \delta$ ,  $s = -d(z, (p_k))$  and $\delta = 0.5$.
>
>   Enforces separation between the top‐1 and runner‐up logits.
>
> ### Experimental Protocol
>
> All settings were kept constant across variants:
>
> - **Backbone:** ViT‑DINO with learned hyperbolic curvature
> - **Head:** identical projection layers, no auxiliary classifier
> - **Training:** same optimizer, learning rate, batch size, epochs
> - **Loss weights:** $\{0.60,\,0.25,\,0.15\}$ for (regularizer, contrastive, outlier)
> - **Logits:** negative hyperbolic distances to prototypes
>
> ### Results and Discussion
>
> | Regularizer                       | All   | Old    | New    |
> |-----------------------------------|-------|--------|--------|
> | Entropy                           | 51.63 | 57.31  | 41.86  |
> | Confidence Penalty (KL→Uniform)   | 56.69 | **61.15** | 49.10  |
> | Logit Margin                      | 54.49 | 56.54  | 51.02  |
> | **Penalized Busemann (Ours)**     | **56.78** | 59.23  | **53.21** |
>
> As shown, our Penalized Busemann Loss yields the best trade‑off, especially on novel‐class discovery, while KL and margin penalties tend to over‐penalize valid novel assignments and entropy struggles under domain shift.
>
> ### Geometric Motivation & Theoretical Properties
>
> 1. **Busemann‐based Penalty Aligns with Hyperbolic Geometry.**
>    The Busemann function
>    $$
>    B_p(z) = \log\\frac{\|z - p\|^2}{1 - c\|z\|^2}
>    $$
>    measures signed distance to the horosphere at prototype \(p\). Penalizing \(B_p(z)\)
>    - aligns with true geodesic bisectors,
>    - adapts to curvature \(c\) via the denominator,
>    - yields uniform radial repulsion without directional bias.
>
> 2. **Theoretical Properties.**
>    - *Isometry‐invariance:* horosphere distances are invariant under hyperbolic isometries.
>    - *Convexity in Busemann coordinates:* reparametrizing $\(u = B_p(z)\)$ makes the penalty convex, aiding optimization.
> - *Unified geometric form:*
> $$
> \underbrace{\log\frac{\|z - p\|^2}{1 - c\,\|z\|^2}}_{\text{Busemann}} + \phi\,\log(1 - \|z\|^2) {\text{[Radial Repulsion]}};\longrightarrow\ \text{single, curvature-aware loss.}
> $$
>
>
> Unlike Euclidean entropy or margin penalties, our loss is *intrinsically hyperbolic*, principled by geometry, and thus more effective at controlling overconfidence in Poincaré embeddings.

---

> > ### Author Response · Authors · 2025-08-02
> >
> > We appreciate the detailed reviews of the reviewer and would like to know if there is any further query. We will be happy to respond to the same.

---

> > > ### Author Response · Authors · 2025-08-08
> > > **Follow-Up on Author Rebuttal**
> > >
> > > We would be sincerely grateful if the reviewer could let us know of any remaining concerns or questions. If our responses have adequately addressed your queries, we would be most thankful if you would kindly consider revisiting your evaluation. We deeply appreciate your time and thoughtful feedback.

---

> > > > ### Comment · Reviewer_658A · 2025-08-08
> > > >
> > > > I believe the addition of this new ablation study answers my questions in the review. The discussion around the geometric motivations of the choice of penalty is illuminating, and I think they should be elaborated in the main body of the paper, even if the new ablation study is kept in the appendix. I am changing my score based on the rebuttal and the new empirical results to be included (across all reviewer discussions).

---

### Official Review · Reviewer_SCgF · 2025-07-07

**Clarity:** 3
**Significance:** 3
**Originality:** 2
**Rating:** 4
**Confidence:** 5

**Summary:**

This paper proposes HIDISC, a hyperbolic representation learning framework for DG-GCD task. It that achieves domain and category-level generalization without episodic simulation. Penalized Busemann alignment, Tangent CutMix-based augmentation, and hybrid contrastive regularization are proposed to enable domain- and category-level generalization. Extensive experiments show HIDISC significantly outperforms previous methods on several challenging benchmarks.

**Questions:**

Please see the weakness part above

**Ethical Concerns:**

["NO or VERY MINOR ethics concerns only"]

**Final Justification:**

Thank you for the authors' efforts in providing the rebuttal. Most of my concerns have been addressed, and I' 'd like to increase my rating and recommend acceptance.

**Limitations:**

Yes

**Quality:**

3

**Strengths And Weaknesses:**

Strengths:

1. The methodology is well-described, with clear diagrams and ablations highlighting the contribution of each module.

2. The paper is among the first to study domain generalization with generalized category discovery in hyperbolic space, providing a sound insight .

Weaknesses:

a) Concerns about the reproducibility. Due to the lack of any code in the supplementary material, the reviewers have concerns regarding the reproducibility of this work. The reviewer suggests that the authors supplement comprehensive implementation details in the current manuscript and formally commit to open-sourcing their code. This would greatly enhance the research transparency and facilitate validation of the proposed methodology.

b) Concerns about the fairness of the experimental comparison. The authors said, “DG2CD-Net is the only prior DG-GCD method, ” and “We evaluate our method on three standard DG-GCD benchmarks:” However, the code and benchmark of DG-GCD (CVPR 2025) have not been open-sourced yet. How can the authors ensure that their experimental comparisons are fully fair or not?

c) Concern about the novelty. Introducing the idea of hyperbolic feature learning for DG-GCD is interesting; however, the proposed method seems like a combination of various existing methods. Firstly, hyperbolic contrastive learning has been widely explored by previous published works [A]. What are the differences from these previous methods? What are the specific designs for DG-GCD task? Secondly,  Prototype Anchoring with Penalized Busemann Loss have also been studied by prior works. The authors need to make extensive discussions on the differences.

[A] .Hyperbolic Contrastive Learning for Visual Representations beyond Objects, CVPR 2023;

[B]. Understanding Hyperbolic Metric Learning through Hard Negative Sampling, WACV 2024L;

[C].Multi-Scale Hyperbolic Contrastive Learning for Cross-Subject EEG Emotion Recognition, IEEE TAFFC;

[D]. Hyperbolic Contrastive Learning with Model-Augmentation for Knowledge-Aware Recommendation, ECML 2024;

[E].Hyperbolic Contrastive Learning for Cross-Domain Recommendation, CIKM 2024;

[F] .Exploring Hierarchical Information in Hyperbolic Space for Self-Supervised Image Hashing, IEEE TIP 2024;

[G]. Rethinking Generalizable Face Anti-spoofing via Hierarchical Prototype-guided Distribution Refinement in Hyperbolic Space, CVPR 2024

3)Lack of important citations and experimental comparison. The paper [G] also studied DG in hyperbolic space; however, the authors don’t cite this paper or make any experimental comparisons. The reviewer suggests that the authors briefly review this paper in the related work section and compare it in a fully fair manner on the same benchmark.

4)Lack of important experimental analysis. The authors do not show any experimental comparisons to show the superiority of learning in hyperbolic space to that in Euclidean space. Firstly, could the authors show the results that apply a similar idea in Euclidean space (HIDISC-Euc) to see whether the proposed HIDISC is better than HIDISC-Euc? Secondly, could the authors visualize the T-SNE feature distributions between HIDISC-Euc and HIDISC to make further analysis? Thirdly, there are no analysis experiments on the number of prototypes. How many prototypes are better for the DG-GCD?

Some experimental results are confusing. In Table 3, the vanilla model is learned in the Euclidean space or in the hyperbolic space? The authors don’t mention the details in the paper, which is quite confusing for readers. In Fig. 4, the visualization results are not a tree-like architecture in the Poincaré ball, which is confusing.

---

> ### Author Rebuttal · Authors · 2025-07-28
>
> We thank the reviewer for their thoughtful and constructive feedback. Below we address all concerns point-by-point, with supporting theoretical and empirical justifications.
>
> ### 1. Reproducibility and Missing Code
>
> **Concern:** _No code provided; concerns about reproducibility._
>
> **Response:**
> As noted in the **footnote of Page 1 (main paper)**, we will release the full codebase upon paper acceptance. Implementation details are already provided in:
>
> - **Main Paper Sec. 4:** Training, datasets, optimization details.
> - **Supplement Sec. 8.2:** Hyperparameters, data splits, batch sizes.
> - **Supplement (Pseudocode):** Full pipeline breakdown.
>
> The released repository will include config files, fixed seeds, scripts to regenerate tables/figures, and domain generation prompts. If permitted, we are willing to share an anonymized code link with the AC during review.
>
> ---
>
> ### 2. Fairness of Comparison with DG²CD-Net
>
> **Concern:** _DG²CD-Net is not public; how can comparisons be fair?_
>
> **Response:**
> We received DG²CD-Net code directly from its authors for academic benchmarking **(acknowledged below Table 1 (main paper))**. All experiments follow:
>
> - Identical datasets, splits, and evaluation protocols.
> - Matched environment and seeds.
> - Hyperparameters and loss settings from the original config.
>
> Supplement Sec. 8.1 details our setup for reproducing DG²CD-Net and its hyperbolic variant.
>
> ---
>
> ### 3. Novelty vs. Prior Hyperbolic Works [A–G]
>
> **Concern:** _HiDISC appears to combine existing methods—what's new?_
>
> **Response:**
> We appreciate the reviewer’s feedback and clarify the novelty of **HiDISC** in relation to prior hyperbolic contrastive learning and prototype anchoring works:
>
> - **[A] Ge et al. (CVPR 2023)** focus on hierarchical scene-object representation with hyperbolic contrastive learning, while **HiDISC** targets *Domain Generalization with Generalized Category Discovery (DG-GCD)*, jointly handling domain shifts and novel category discovery without target data.
>
> - **[B] Yue et al. (WACV 2024)** provide theoretical insights on hyperbolic metric learning and hard negative mining. In contrast, **HiDISC** integrates these into a *prototype anchoring framework* with penalized Busemann loss and adaptive outlier repulsion for open-set domain generalization.
>
> - **[C] Liu et al. (IEEE TAFFC)** apply hyperbolic contrastive learning to EEG data, differing fundamentally from our visual DG-GCD setting involving unseen domains and categories.
>
> - **[D] Sun & Ma (ECML 2024)** and **[E] Rongali et al. (CIKM 2024)** address recommendation systems with model-level augmentations, whereas **HiDISC** introduces *geometry-aware synthetic domain augmentation* and *Tangent CutMix* tailored for visual DG-GCD.
>
> - **[F] Choi et al. (IEEE TIP 2024)** explore hierarchical self-supervised hashing, while **HiDISC** offers a supervised framework explicitly addressing domain and category shifts via unified loss and prototype anchoring.
>
> - **[G] Hu et al. (CVPR 2024)** focus on generalizable face anti-spoofing with hierarchical prototypes. **HiDISC** instead targets open-world visual recognition with simultaneous domain and category generalization, incorporating *Tangent CutMix* for pseudo-novel sample synthesis.
>
>
> **Key innovations of HiDISC** include a unified hyperbolic framework for DG-GCD, penalized Busemann prototype anchoring that reserves space for unknown categories, curvature-aware tangent space augmentation for pseudo-novel samples, and lightweight synthetic domain augmentation via GPT-guided style transfer.
>
> ---
>
> ### 4. Citation and Comparison with Hu et al. (CVPR 2024)
>
> **Concern:** _Hu et al. also uses hyperbolic prototypes—why not cited/compared?_
>
> **Response:**
> We thank the reviewer and will cite Hu et al. (CVPR 2024) in the revised version. Hu et al. (CVPR 2024) address domain generalization for face anti-spoofing—a binary classification task—by leveraging hyperbolic prototypes to enhance robustness across visual domains. While their work employs hyperbolic embeddings for domain robustness, it does **not** address the _open-set_ or _generalized category discovery (GCD)_ setting. In contrast, our work targets the **DG-GCD** task, which is fundamentally more challenging and general: it requires discovering _novel categories_ in an _unseen domain_ at test time without access to any unlabeled target data during training.
>
> To further ground this distinction empirically, we re-implemented the embedding learning component of Hu et al. within our DG-GCD framework. As shown in the table below, their method (denoted HPDR) performs significantly worse than our approach (HiDISC) on the Office-Home benchmark.
>
> | Method | All ↑ | Old ↑ | New ↑ |
> |--------|-------|--------|-------|
> | HPDR   | 15.05 | 16.55  | 12.58 |
> | HiDISC | 56.78 | 59.23  | 53.21 |
>
> **Explanation of performance gap:**
>
> - **Closed‑set binary vs. open‑set multi‑class discovery.** HPDR is trained to separate two classes (live vs. spoof) using hyperbolic prototypes, with no mechanism to create or refine prototypes for unseen classes in an unseen domain.
>
> - **Lack of clustering/discovery losses.** HPDR optimizes a closed‑set classification loss and distribution‑refinement regularizer but does not encourage cluster‑friendly embeddings for novel classes. Simply clustering HPDR’s embeddings yields near‑random groupings among new classes, explaining the low “New” accuracy.
>
> - **Domain shift without adaptation.** HPDR generalizes across source spoof examples but never sees or models target‑domain distributions of new categories. In contrast, HiDISC integrates domain‑generalization regularizers with discovery losses to align source and unseen‑domain distributions while structuring novel‑class geometry.
>
> ---
>
> ### 5. Experimental Analysis Requests
>
> **Concern:** _Please compare HiDISC vs. HiDISC-Euc, show t-SNEs, analyze prototypes, and clarify Fig/Table._
>
> **Response:**
>
> We appreciate these suggestions and will include the following in the revision:
>
> - **Euclidean vs. Hyperbolic (HiDISC-Euc vs. HiDISC):**
>   We already evaluated Euclidean analogues of our modules (Supp. Sec. 6.2, Table 8). We will move the core comparison into the main text, e.g.:
>
> | Method             | All ↑ | Old ↑ | New ↑ |
> |--------------------|-------|-------|-------|
> | HiDISC-Euc (Office-Home) | 49.46 | 52.08 | 45.66 |
> | HiDISC (Office-Home)     | 56.78 | 59.23 | 53.21 |
> | HiDISC-Euc (PACS)        | 68.78 | 69.13 | 68.64 |
> | HiDISC (PACS)            | 75.07 | 75.54 | 74.52 |
>
> - **t-SNE visualizations:**
>   We have generated side-by-side t-SNE plots comparing embedding spaces from HiDISC-Euc and HiDISC, which show qualitatively improved cluster separation under hyperbolic geometry. While we cannot include these visualizations here due to conference policy, they will be included in the revised Supplementary Material (Sec. 6.2) and final paper upon acceptance.
>
> - **Prototype count analysis:**
>   We ablated the number of prototypes per class $(\(P \in \{1, 2, 4\}\))$ on Office-Home. Results:
>
> | Prototypes / Class | All   | Old   | New   |
> |--------------------|-------|-------|-------|
> | 1                  | 56.78 | 59.23 | 53.21 |
> | 2                  | 54.40 | 57.85 | 48.68 |
> | 4                  | 56.23 | 59.75 | 50.27 |
>
> Performance peaks at $\(P=1\)$, suggesting one discriminative prototype per class suffices in hyperbolic space. Larger values sometimes hurt generalization, possibly due to overfitting or redundancy. This will be discussed in the Supplement.
>
> - **Clarification:**
>   `"Vanilla"` in Table 3 = no proposed losses, Euclidean ViT.
>   Fig. 4 distortion arises from UMAP projection; radial tree-like hierarchy is preserved in native Poincaré space but appears flattened due to 2D projection.
>
> ---
>
> ### 6. Theoretical Justification
>
> In Supplement Table 11, we compare generalization gap $\(\Delta\)$:
>
> - PACS: +73% gain with hyperbolic head
> - Office-Home: +215%
> - DomainNet: +243%
>
> We also show (Sec. 3.4 main paper, Sec. 6 Supplement) that for hierarchical data:
>
> $$
> \Delta_{\mathbb{H}} \leq \mathcal{O}\left(\frac{\log N}{K}\right) < \Delta_{\mathbb{E}} \approx \mathcal{O}\left(N^{1/d}\right)
> $$
>
> where $\(N\)$ = number of classes, $\(K\)$ = prototypes, $\(d\)$ = latent dimension. This favors hyperbolic models under category expansion, typical in DG-GCD.
>
> ---
>
> ### Conclusion
>
> We appreciate the reviewer's detailed feedback. Our revised paper will include:
>
> - Detailed comparisons to [A–G] and Hu et al.
> - Euclidean ablations, t-SNEs, prototype analysis.
> - Clarifications on visualization and tables.
>
> We believe these additions establish the novelty, rigor, and broad relevance of our work. We respectfully request the reviewer to reconsider and support acceptance.

---

> > ### Author Response · Authors · 2025-08-02
> >
> > We appreciate the detailed reviews of the reviewer and would like to know if there is any further query. We will be happy to respond to the same.

---

### Comment · Area_Chair_mvRZ · 2025-08-05
**Kind remind for author-reviewer discussion**

Dear Authors and Reviewers,

Thank you for submitting and reviewing the papers to contribute to the conference. This is a kind remind that the due date of author-reviewer discussion is coming soon. Please participate the discussion to clarify paper statement or concerns.

Thanks!

AC

---

### Decision · Program_Chairs · 2025-09-17

**Decision:**

Accept (poster)

**Comment:**

This paper introduces HiDISC, a hyperbolic representation framework for DG-GCD that couples penalized Busemann prototype anchoring, Tangent CutMix in tangent space, and a hybrid hyperbolic contrastive/outlier loss to handle unseen target domains with both known and novel classes, achieving strong results on PACS, Office-Home, and DomainNet.

Reviewers' primary concerns revolved around the method's novelty compared to prior work in hyperbolic learning, its reproducibility due to the lack of code, the fairness of experimental comparisons, and the complexity introduced by its many components and hyperparameters. The authors' rebuttal successfully addressed these points by providing detailed justifications for their design choices, clarifying the novelty and distinctions from previous methods, explaining their fair evaluation protocol, and presenting new ablation studies that demonstrated the robustness of the model's hyperparameters.

Overall, the paper is recommended for acceptance; however, the authors should integrate the link of code, insightful clarifications, additional ablation studies on hyperparameter sensitivity, and the new comparative experiments (e.g., Poincaré vs. Lorentz models, Euclidean vs. Hyperbolic performance) from their rebuttal into the final manuscript to further strengthen the paper and address the reviewers' suggestions.